# The Determinants of Consumers' E-Waste Recycling Behavior through the Lens of Extended Theory of Planned Behavior

**Nur Shafeera Mohamad [1], Ai Chin Thoo [1,\*] and Hon Tat Huam [2]**

[1] Azman Hashim International Business School, Universiti Teknologi Malaysia, Skudai 81310, Malaysia; nshafeera2@graduate.utm.my
[2] Putra Business School, Universiti Putra Malaysia, Serdang 43400, Malaysia; huam@putrabs.edu.my
\* Correspondence: acthoo@utm.my

**Abstract:** Cutting-edge technological advancements have turned many electronic devices into waste within a short time of usage. Electronic waste (e-waste) has become a global problem. Due to the adverse impact of e-waste on the environment and human health, a regulatory system for effective collection and treatment of e-waste disposed of by the community and business sectors is in dire need. In an attempt to address the setbacks and cope with e-waste issues, this study determined factors that influence e-waste recycling intentions (ERIs) and e-waste recycling behavior (ERB) among consumers in Malaysia through the lens of an extended theory of planned behavior (TPB) model. Via an online survey, 159 questionnaires were collected from targeted respondents in Malaysia aged 18 years and above identified using the purposive sampling method. The structural equation modeling (SEM) approach was deployed for data analysis. As a result, moral obligation (MO) emerged as the most significant factor toward ERI, followed by perceived convenience (PC). Next, ERI displayed a significant effect on ERB. The multi-group analysis (MGA) outcomes revealed significant group differences in education variables, signifying that the lower/middle education group was more easily influenced to perform ERB than the higher education group. Essentially, this study contributes to many aspects especially in enhancing the awareness of e-waste issues and emphasizing the broad knowledge about e-waste recycling. This study provides practical implications for the government, the policymakers and all stakeholders, including consumers, non-government agencies, collectors, retailers, and recycling facilities. The study outcomes may be considered when formulating laws and regulations to enhance e-waste recycling efforts that guarantee a sustainable ecosystem in the future.

**Keywords:** electronic waste (e-waste); theory of planned behavior (TPB); e-waste recycling behavior (ERB); awareness of environmental consequences (AEC); perceived convenience (PC); moral obligation (MO)

## 1. Introduction

Over the last two decades, the average life cycle of electrical and electronic goods has reduced due to the rapid technological progress in line with the ever-escalating consumer demand [1]. The increasing usage of electronic devices has led to enormous volumes of electronic waste (e-waste) [2]. In fact, e-waste is one of the world's fastest emerging waste categories, which records 3–5% on an annual basis [3]. Based on a study published by the United Nations University's Global E-waste Monitor 2020, 53.6 million metric tons (Mt) of e-waste were generated in 2019 [4,5]. Concurrently, Asian countries generate nearly half of this enormous figure—24.9 Mt. Forti [6] estimated that 74.7 million Mt of e-waste would be produced by 2030. If significant measures are not taken, this number is expected to rise to 120 Mt by 2050. Unfortunately, only 20% of this tremendous amount of e-waste was recycled in some useful sense [7–9]. The remaining 80% were usually disposed of at the landfill, thus causing harm to the environment [9]. Turning to Malaysia, it yielded 364 kilotons of e-waste or 11.1 kg of e-waste per capita in 2020 [10]. The Malaysian government statistics,

nonetheless, reported that about 25% of e-waste in this country is recycled, while the remaining e-waste improperly recycled was worth about MYR 3 billion [11].

Yong [12] highlighted that e-waste generation is now a significant issue across all countries in the world, mainly because the harmful elements and chemical substances can cause adverse effects on both the environment and human health. These hazardous substances (i.e., lead, arsenic, cadmium, mercury, cathode ray tubes, chromium, and poly-brominated biphenyls), if mismanaged, are detrimental to the environment and human health [4,13–19]. In addition, e-waste releases greenhouse gases and ozone-depleting substances. For instance, both refrigerators and air-conditioners contain chlorofluorocarbons (CFCs) (Freon) gases that significantly contribute to global warming and ozone depletion. As ozone depletion increases ultraviolet (UV) radiation to the surface of the Earth, many may succumb to skin cancer [20]. On the other hand, e-waste is highly valued as secondary raw materials, such as gold, silver, platinum and palladium, iron, copper, aluminum, and plastics, which can be extracted and sold [21,22]. According to Widmer [22], the recovery of these materials from e-waste projects a profitable venture. Since a huge number of precious metals can be found in e-waste, the recovery of e-waste signifies a lucrative economic advantage as it limits spending on costly and scarce resources needed to generate new electronic goods [3].

Instead of being thrown away, e-waste should be reused, resold, recovered, remanufactured, recycled, or disposed of via a reverse logistics process. Electrical and electronic equipment (EEE) reverse logistics has become the focal point for lawmakers, researchers, and producers [23]. The rising amount of e-waste in huge volumes, along with poor environment-friendly recycling structures, has been reckoned as a serious issue [24]. However, the literature on reverse logistics merely concentrates on the recycling theory designed for companies or manufacturers, while omitting the link between reverse logistics system and consumer behavior [25]. Notably, the reverse logistics process cannot be operated effectively without consumer participation because they are the initial connection in the entire supply chain [26]. The absence of consumer participation in logistics e-waste recycling can lead to e-waste being disposed of either by incinerating or by throwing it at landfills, which is detrimental to the environment [25]. Moreover, the literature insufficiently depicts the intention of consumers to sell or recycle, as well as their participation in e-waste management across developing countries [23]. As such, this study outlined the key determinants of e-waste recycling intentions (ERIs) and e-waste recycling behavior (ERB) due to the scarcity of studies related to reverse logistics and recycling rates in Malaysia.

According to Zhang [27], large volumes of e-waste have flooded to unauthorized recycling facilities, where the waste is processed with rough refining and non-environment-friendly methods. They added that e-waste collection is hindered due to the unwillingness among consumers to send e-waste to authorized disposal facilities. Haron [2] claimed that the other obstacles are due to the existence of unauthorized waste collectors, as well as a lack of awareness and knowledge among customers, retailers, and producers. Free Malaysia Today [28] stated that such activity stems from illegal facilities located near residential areas, which could further lead to pollution that causes significant health issues for the residents living there.

Therefore, this present study bridged the gaps by assessing factors that influence e-waste recycling participation and awareness of e-waste issues through the lens of the theory of planned behavior (TPB). However, the TPB within the recycling context has limitations because it is complicated for the model to predict or determine behavior that is not from personal desire and decision. Recycling behavior involves external resources and expertise [29]. For example, the key factors for unwillingness to recycle e-waste in developing countries are a shortage of recycling facilities [30], non-strategic e-waste disposal centers that are located far from residential areas [31], and a lack of knowledge about the deleterious impacts of e-waste on the environment and human health [32].

To sum up, everything that has been stated so far, regarding the academic gaps, there are still limited studies investigating the connections between reverse logistics and

consumer context in Malaysia because most of the previous literature focused on reverse logistics with companies or manufacturers' contexts. In addition, there is still insufficient information from the previous research literature about the intention of consumers to sell or recycle as well as the participation of consumers in e-waste management in developing countries, especially in Malaysia. Moreover, in terms of a theoretical gap, even though the TPB model is suitable for predicting recycling behavior, there are still limitations in the TPB model. This is because recycling behavior involves personal desire and decision as well as external resources and expertise, which is out of the individual's control. Lastly, regarding the practical gaps, there are vast volumes of e-wastes that have flooded to unauthorized recycling facilities which use inappropriate recycling methods and incinerate some valuable parts of e-waste that can be resold. The reason for this circumstance is that prior research discovered that Malaysians have limited knowledge of e-waste and its repercussions [4,33]. Thus, Malaysians were reported to be uninformed about proper e-waste disposal.

Consequently, to bridge the gaps, there is a need for an extensive understanding of consumers' behavior concerning e-waste and the aspects that will improve consumer intentions to participate in e-waste recycling. To address these setbacks, an extended TPB model was used in this study to assess recycling intention and behavior among consumers in Malaysia by embedding extra variables: awareness of environmental consequences (AEC), perceived convenience (PC), and moral obligation (MO).

In addition, this paper is the first study that included moral obligation (MO) in the extended TPB to investigate consumers' e-waste recycling behavior in Malaysia since the importance of MO as a predictor of an individual's pro-environmental behavior is undervalued. Hence, the impact of moral obligation should be highlighted in behavioral theories such as the TPB since previous studies believed that moral obligation could be more influential than attitude [34,35] and subjective norms [36] in the study of pro-environmental behavior. Evidence of this can be found in Razali's [37] study which found that moral obligation is the most significant factor in determining household waste separation behavior in Malaysia. Furthermore, Juliana [38] and Sulaiman and Chan [39] also discovered that moral obligations also had a significant effect on recycling behavior in Malaysia. The findings present convincing evidence that moral obligation plays an important role in Malaysian consumers' behavior. According to Juliana [38], people in Malaysia were found to have a sense of guilt if they did not engage in recycling behaviors, and failing to recycle would directly contradict their values and principles. Thus, this study wanted to explore the impact of MO on consumers' recycling behavior, specifically in the e-waste recycling context.

Moreover, no studies exist examining the impact of consumers' socio-demographic factors in performing e-waste recycling behavior in Malaysia. Thus, this study conducted a multi-group analysis (MGA) in order to investigate the significant disparities that exist across the various groups by employing socio-demographic factors such as gender, education level, and income level in performing e-waste recycling in Malaysia. The implication of the findings is beneficial for the government and all stakeholders, including generators or consumers, non-government agencies, collectors, retailers, and recycling facilities, to promote and encourage e-waste recycling among consumers in Malaysia. Moreover, the study outcomes may be considered when formulating laws and regulations to enhance e-waste recycling efforts that guarantee a sustainable ecosystem in the future.

## 2. Literature Review and Hypothesis Development

### 2.1. Reverse Logistics

Reverse logistics refers to the system associated with recycling or transferring products from their typical ultimate destination to achieve proper disposal and to capture value [40], such as gold, silver, and copper [41]. In order to recapture the value of e-waste, old or end-of-life electronic appliances should be recycled and remanufactured, whereby the e-waste would undergo a series of processes to extract valuable material from the discarded products or components to be re-used in future products, while remanufacture is restoring

used products that require complete disassembly of products before proceeding with extensive testing, restoration, and replacement [42].

According to Khor and Udin [43], reverse logistics is a lucrative business because it extends the internal management of the environment, besides being an alternative for organizations to gain environmental reputation benefits, reduce inventory purchasing costs, and improve secondary market as the demand of goods is affected by technological obsolescence and life cycle phases. Ab Halim Nik Abdullah and Yaakub [44] suggested that reverse logistics is more intricate than forward logistics—it is easier to forecast future sales in forward logistics than in reverse logistics. This is because reverse logistics starts with customers' action and involves varying supply chain members. Similarly, Kochan [45] asserted that the involvement of customers is vital in reverse logistics as they act as the initial link in the entire supply chain. Reverse logistics may extend the lifespan of a product by implementing an efficient process through a reverse supply chain, including the flow of goods, components, and information from the consumption point to the start point [46].

Alnuwairan [47] claimed that the collaboration between manufacturers and third parties in reverse logistics activities can minimize costs of procurement, inventory holding, transportation, and disposal if the activities are well-managed by them (manufacturers and third parties). Reverse logistics offers vast advantages to customers, organizations, and the environment. For example, activities that promote reusing, recycling, and reducing the amount of e-waste cause organizations to become more environmentally efficient and increase their goodwill among customers. Such reverse logistics activities also protect consumers from exposure to hazardous elements [44].

### 2.2. Theoretical Background

It is crucial to identify the determining factors of consumers' intention to return their e-waste to the manufacturers or local authorities so that the manufacturers can implement reverse logistics on the e-waste products [31]. Both the theory of reasoned action (TRA) [48] and the TPB [49] have been widely applied in attitude-behavioral studies [50]. As such, prior scholars have highlighted several criticisms about the TRA. One of the critics suggested that the TRA is only applicable to predict totally within volitional control of one's behavior; thus prediction is weakened when implemented for non-volitional control of behavior [45,50]. Similarly, Liska [51] asserted that the inability to conduct specific behavior is due to inadequate opportunities, knowledge, skills, and time. These situations weaken the ability of the TRA to evaluate one's behavior effectively [52]. To overcome the limitations, Ajzen [49] introduced the TPB—an extension of the TRA to deal with one's inadequate volitional control toward the target behavior [49,53]. The TPB has been one of the most extensively used and influential models of attitude–behavior relationship studies for the past two decades [50].

In this present study, the TPB was deployed to assess factors that influenced consumer behavior toward logistics e-waste recycling in Malaysia. Wang [54] defined the TPB as an attitude–behavior relationship, in which human behavior derives from planned behavior and behavioral intentions act as a significant determinant in recycling behavior. Hence, the TPB can be used to assess the effect of using different factors on behavioral decisions systematically. Pashaei and Shahmoradi [55] stated that behavioral intention is the most significant determinant of one's behavior in the TPB.

Kumar [23] found that the TPB framework can be well applied in the education, health, environment, consumer behavior, and technology domains despite certain drawbacks. For instance, Lee et al. (2010) used the TPB to examine the intentions of educators to use technology in classrooms. Alam and Sayuti [56] applied the TPB as a theoretical framework to extend past studies on the behavior of Malaysians purchasing halal food. Meanwhile, Greaves [57] deployed the TPB to investigate intentions to improve organizational environmental behavior as this theory can be used to explore how employee behavior can be utilized to achieve environmental improvements. Jahilian and Emdadi [58] employed the TPB to evaluate factors that contributed to routine Pap-smear tests amongst women aged

20–70 years old. In a study conducted by Groot and Steg [59], the TPB was applied to assess the purpose for which residents in Groningen (the Netherlands) used the park-and-ride facility (transferium). Next, Kaveh [60] deployed the TPB to examine the impact of a school-based nutrition education intervention to improve nutritional behavior among adolescents. Cheon [61] applied the TPB as a conceptual framework to identify college students' perceptions and needs for mobile learning.

Return behavior is subject to many non-psychological limitations, including access to appropriate networks not being under the control of other person(s) [62]. Thus, the TBP is indeed suitable to predict pro-environmental actions, such as returning e-waste [31,63]. Strydom [29] used the TPB to assess recycling behavior in South African urban households, while Pakpour [64] used the TPB to determine factors linked to household waste behavior in Iran. Davis and Morgan [65] deployed the TPB to determine recycling and waste minimization behavior in Bristol City (UK). The TPB is generally applied to examine factors affecting ERB, inclusive of environmental awareness, recycling attitudes and practices, demographic factors, subjective norms (SNs), individual norms, responsibility awareness, legislation and regulations, and information publicity [54]. The three factors of consumers' intention in the TPB that can lead one to perform specific behavior are attitude, SN, and perceived behavioral control (PBC).

In the TPB, attitude toward the behavior refers to the extent to which one has a positive or negative opinion or evaluation of conducting the target behavior [23,49,61]. Attitude denotes behavioral beliefs, which are beliefs in possible consequences or other behavioral characteristics [66]. Kaiser [34] claimed that attitude involves not only assessing a particular result but also calculating the probability of the outcome and providing great facts as experiential knowledge—a fundamental requirement for every kind of attitude. Prior studies proposed that the impact of attitude on environmental behavior is significantly positive (see [24,31,45,57,63,67–69]). For instance, Zhang [27] found that attitude had a significantly positive impact on one's willingness to execute online recycling behavior. Wan [52] and Tonglet [70] reported that attitude emerged as the most significant determinant of recycling intention. Wan [52] asserted that attitude toward recycling can be achieved by promoting recycling educational programs or campaigns to the community. The programs should highlight the benefits of recycling and how recycling practices can shift the attitude of individuals toward a better environment. Tonglet [70] suggested that recycling attitudes are shaped by gaining adequate incentives, having easy access to disposal centers, possessing good knowledge about recycling, and no dissuading physical recycling issues. Greaves [57] noted that a recycling attitude can be formed by performing a favorable assessment toward recycling, such as assuming that waste disposal is responsible, conscientious, and convenient to perform.

Subjective norm (SN) refers to beliefs, perceptions, pressures, and factors that can be either positive or negative in one's social group [23,49]. Singh [71] defined social pressure as a mixture of injunction and descriptive norms that signify social networks and beliefs of the immediate surrounding societies toward certain behavior. According to Wan [52], SN is one's determination to fulfill the expectations of other(s). Since recycling behavior is more likely to include elements of moral and social responsibility, SN is a significant predictor of recycling behavior [45]. Next, Wang [24] stated that SN in the recycling context may be formed by implementing suitable rules and regulations on e-waste recycling so that the behavior and the mentality of the surrounding people can influence the eagerness of the residents to engage in the recycling process. Dixit and Badgaiyan [31] claimed that one will not engage in the target behavior if pro-environmental activities (e.g., recycling) in society are not publicly adopted despite the individual having strong personal values about environmental issues. Tonglet [70] claimed that recycling behavior can be achieved through the impact of social pressure encountered by people, along with personal moral beliefs and social responsibility. Essentially, many past studies reported that SN has a positive impact on recycling behavior (see [23,52,54,68]).

Perceived behavioral control (PBC) denotes the perception and the understanding of one's capability based on his or her past experience and perceived challenges or difficulties to perform the target behavior [24]. Echegaray and Hansstein [63] asserted that PBC reflects how well a person thinks he or she can conquer the challenges and take advantage of the facilitators when executing an action. PBC is composed of two parts: (1) self-efficacy or the ease or complexity of dealing with the performance of the behavior and (2) perceived controllability or the belief that one has control over the behavior [57,66]. In addition, PBC not only predicts behavioral intention but also predicts behavior and intention [24,64]. According to Echegaray and Hansstein [63], the existence of nearby disposal facilities could influence PBC in recycling so that the consumers believe they can save their time when performing e-waste recycling activities. On the contrary, Wang [24] found that PBC is primarily evaluated by the experience of recycling. Hence, someone with experience in recycling would be more interested in participating in further recycling activities when compared to someone without recycling experience. Past studies have mostly revealed the significant effect of PBC on pro-environmental behavior, such as recycling (see [52,64,68,72]). In conclusion, PBC is a significant indicator of recycling behavior because PBC refers to one's confidence in the likelihood of executing certain behavior despite any external or internal limitation. Someone who is confident to perform recycling activity is more likely to express recycling behavior, compared to someone who feels they have little control over present or future setbacks [70]. Thus, the following hypotheses were developed:

**Hypothesis 1 (H1).** *Attitude has a significantly positive relationship with e-waste recycling intention (ERI).*

**Hypothesis 2 (H2)**. *Subjective norm (SN) has a significantly positive relationship with e-waste recycling intention (ERI).*

**Hypothesis 3 (H3).** *Perceived behavioral control (PBC) has a significantly positive relationship with e-waste recycling intention (ERI).*

*2.3. Extending the Theory of Planned Behavior*

The TPB model in the recycling context has several limitations, mainly because it is complicated for the model to predict or determine behavior that does not stem from personal desire and decision. Recycling behavior involves external resources and expertise [29]. A significant factor for one's unwillingness to recycle e-waste across developing countries is due to the shortage of recycling facilities [30] or non-strategic recycling facilities that are far from residential areas. This return behavior derives from MO [31]. In order to overcome the limitations, the TPB model has been extended to enhance its predictive efficacy. In fact, many have attempted to adapt, change, and expand the TPB for better predictability. According to Kumar [23], many researchers have used the TPB model by adding some factors to influence consumers' behavior, such as moral norms, sense of duty, convenience, and infrastructure, to explore return or recycling behavior. Scholars have also included other variables in the TPB model, such as demographic variables and economic benefits from recycling [54]. Ling [73] stated that many researchers have extended the TPB to enhance its overall percentage of variation in recycling behavior. Ajzen [49] claimed that scholars can improve the explanatory power of the TPB by extending the model.

In this study, several key components were included in the TPB model as predictors of e-waste recycling—AEC, PC, and MO—to assess factors that affect ERI and ERB among consumers in Malaysia. Figure 1 shows the relationships of ERI with attitude, SN, PBC, AEC, PC, MO, and ERB.

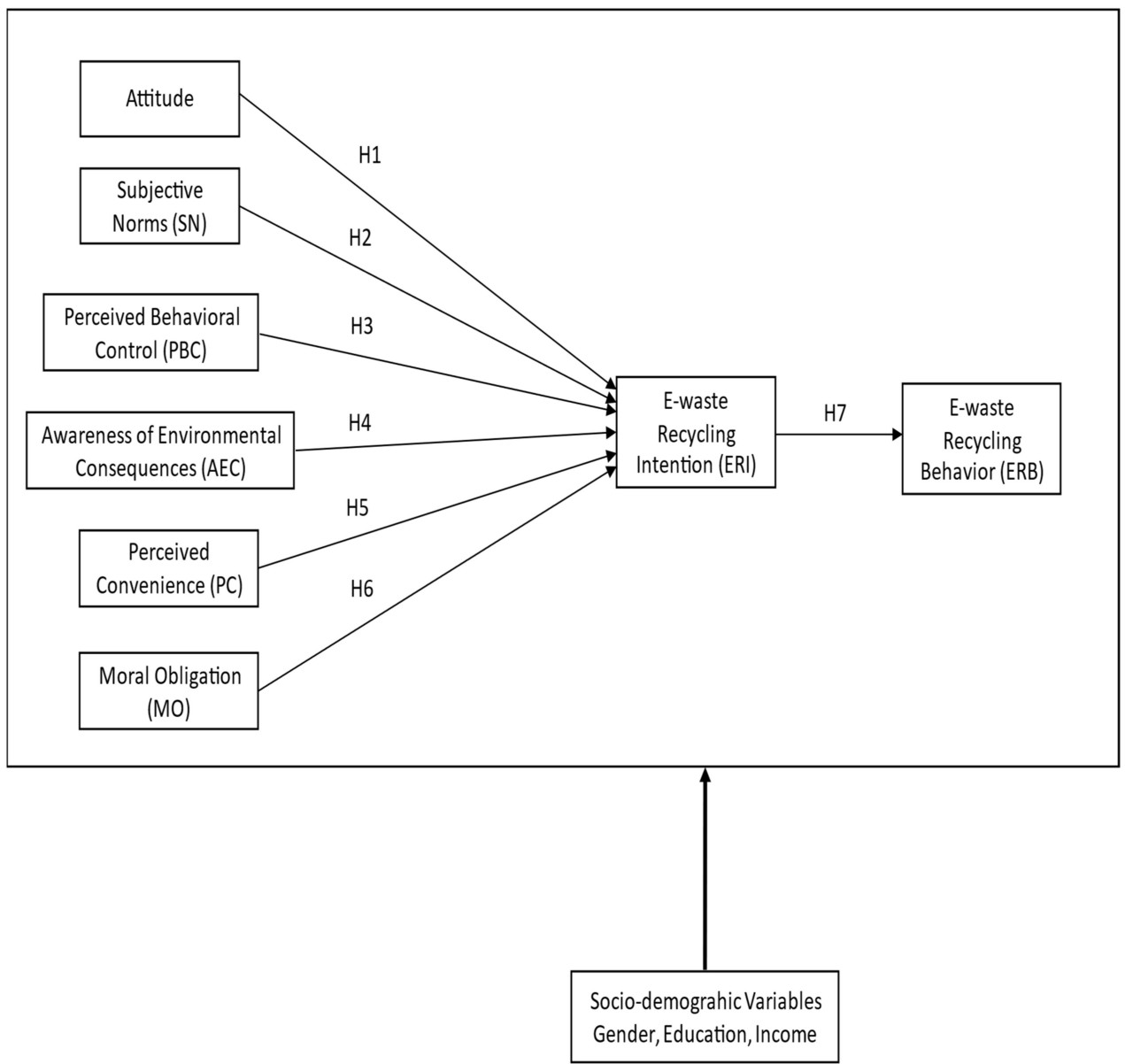

**Figure 1.** Research framework.

### 2.3.1. Awareness of Environmental Consequences and E-Waste Recycling Intention

Awareness of environmental consequences (AEC) refers to one's knowledge and understanding of the environmental effects of performing certain behavior [45]. Saphores [74] and Nnorom [75] proposed that consumers are willing to spend more to buy green electronic equipment when they are aware of the significance of environment conservation. In fact, AEC is viewed as one of the significant motivators for recycling intention in vast empirical studies [45]. Tonglet [70] claimed that AEC potentially triggers the consumers' e-waste recycling behavioral intention. According to Wan [52], AEC is the concept in which a person conscious of the environmental implications might affect his or her ERI. Hence, environmental concern has been depicted as one of the main drivers of recycling behavior in many empirical studies [45]. Studies have revealed that AEC can positively predict recycling intentions (see [25,32,52,74,76]). Afroz [77], for instance, found that 65% of consumers were concerned with environmental impacts when spending on electronic products, signifying awareness among consumers about the adverse environmental impact of electronic products. In a similar vein, Akhtar [32] revealed that more than half of their

respondents (56%) were aware that EEE had caused environmental and human health issues. Thus, this present study looked into the possibility of environmental awareness as a concept that might affect one's consciousness of the environmental impacts of his or her ERB. While assessing the intention of consumers to recycle e-waste, AEC was embedded as a crucial factor in the conceptual framework. As such, the following hypothesis was proposed in light of AEC:

**Hypothesis 4 (H4).** *Awareness of environmental consequences (AEC) has a significantly positive relationship with e-waste recycling intention (ERI).*

2.3.2. Perceived Convenience and E-Waste Recycling Intention

Perceived convenience (PC) denotes the concept of one feeling that the availability of time to remove, sort, and store e-waste has an impact on his or her ERB [52]. A major factor in one's unwillingness to recycle e-waste across developing countries is due to the shortage of recycling facilities which leads to inconvenience for consumers to recycle their e-waste [30,78]. Saphores [79] suggested that a strategic and convenient recycling center location can significantly increase the recycling intention among consumers due to low time consumption and cost. In the context of Malaysia, Mohd Sharif and Soo [25] found that consumers were reluctant to participate in e-waste recycling because the facilities for e-waste recovery provided by the government are primarily focused on industrial waste. Therefore, consumers would need to spend extra time to find out the location of e-waste facilities. Seemingly, the recycling intentions and behavior of consumers can be enhanced by providing adequate physical proximity to recycling bins and curbside collection so that more consumers could participate in recycling activities [30,70]. According to Kochan [45], participation of consumers in recycling may be increased by preparing a closer location of disposal recipients, reducing the difficulty in collecting and processing recyclable waste, as well as providing various recycling collection programs and courses (e.g., recycling collection events and curbside collection). Prior studies demonstrated that convenience is a significant factor in motivating consumers' intention to recycle (see [2,25,27,30,76,79]). For instance, Zhang [27] highlighted that the participation of ERI among consumers increased when more e-waste recycling facilities (e.g., e-waste recycling bins) were established across urban areas in China. This is because, as the recycling facilities offered the consumers a more convenient way to drop their e-waste, they did not need to travel too far to recycle their e-waste. Following the above discussion, this study examined if individuals would perform e-waste recycling action if there was time availability, if it was low in cost, and if there were recycling facilities nearby. Hence, the following was hypothesized:

**Hypothesis 5 (H5).** *Perceived convenience (PC) has a significantly positive relationship with e-waste recycling intention (ERI).*

2.3.3. Moral Obligation and E-Waste Recycling Intention

Moral obligation (MO) thoughts can be related to the self-assigned obligation of a person, such as when one is conscious of making a decision to perform certain behavior [23]. Chen and Tung [80] acknowledged the importance of MO when evaluating behavioral intention. Similarly, Godin [81] indicated that the TPB considers the importance of internalized values, such as MO, throughout the production of internal motivation for the execution of target behavior. Inclusion of MO in the TPB framework can enhance the predictability of the TPB model [82]. According to Thøgersen [83], returning e-waste may be costly because consumers need to go to the disposal center to deposit their e-waste, and the motivation to perform recycling behavior could be intrinsic in the absence of an incentive. Such behavior is defined as moral rather than rational, as it derives from moral norms [31]. Tonglet [70] discovered that an individual who perceives it is necessary to recycle or vice versa is likely to include personal norms in the decision-making process. Bamberg and Möser [84] described environmental awareness as a cognitive prerequisite for

the formation of MO. Numerous past studies have reported a significant link between MO and consumers' intention or behavior (see [35,45,64,70,72,85]). For instance, Pakpour [64] found that MO had the highest predictor of household waste behavior among the other TPB predictors. Meanwhile, Botetzagias [72] revealed that MO displayed a greater impact on recycling intention than attitude. The study found that the participation of society in recycling could be increased through programs that enhance the MO of individuals to conduct recycling practices. Based on the substantial number of evidence concerning the predicted relationship between MO and recycling intentions, this study included MO as a key factor in the conceptual framework. Thus, the following hypothesis was proposed:

**Hypothesis 6 (H6).** *Moral obligation (MO) has a significantly positive relationship with e-waste recycling intention (ERI).*

2.3.4. E-Waste Recycling Intention and E-Waste Recycling Behavior

Intentions refer to a scenario when an individual is instructed to execute specific behavior or to achieve certain results [31]. Defined as a good predictor of behavior, intentions yield the strongest reliable predictor of behavior if a suitable measure of intention can be achieved [63,86–88]. One's intention acts as a primary determinant in deciding if certain behavior is performed [45,73]. Behavioral intention indicates how willingly one wishes to contribute to performing the target behavior [50,86]. The TPB illustrates that behavioral intention acts as a function of three elements, namely attitude, SN, and PBC [63]. Recycling behavioral intention is defined as one's willingness to convert waste into a valuable resource [89]. Kochan [45] defined ERI as one's intent to recycle e-waste, drop off e-waste at a nearby collection point, and return e-waste to the collectors or manufacturers. According to Nguyen [67], the behavioral intention of consumers to participate in e-waste recycling activities is one of the key aspects of e-waste management strategy. This means that the performance of recycling activities depends highly on the involvement of consumers. Several past studies have identified that recycling intention has a positive impact on recycling behavior (see [35,64,68,76,84,88]). For example, Poškus [90] reported that intention is the strongest determinant of recycling behavior. To assess the link between ERI and ERB, the following hypothesis was proposed:

**Hypothesis 7 (H7).** *E-waste recycling intention (ERI) has a significantly positive relationship with e-waste recycling behavior (ERB).*

*2.4. The Effect of Socio-Demographic Factors (Gender, Education Level, and Income) on the Model*

Many incontrovertible studies have established that socio-demographic factors have a significant influence on predicting consumers' environmental behavior [91]. According to Sidique [30], socioeconomic variables (e.g., age, education level, and family size) substantially affect recycling attitude and behavior. Wang [92] showed that socio-demographic characteristics (e.g., education, income, and family size) had little impact on consumers' intention to recycle e-waste. Past studies (see [67,91]) revealed that men were more willing to recycle than women, while Johnson [93] found that women were more inclined to turn their intention into behavior in the recycling context. Gender, according to do Valle [94], is neither statistically nor significantly linked with ERI. Diamantopoulos [95] reported that environmental knowledge, attitude, and behavior were significantly related to education. Xu [91] suggested that only a handful of studies have assessed the impact of education intervention on waste separation behavior.

Babaei [96] demonstrated that demographic variables (e.g., age, education level, gender, and occupation) had an impact on one's knowledge, attitude, and habits regarding solid waste recycling. Botetzagias [72] added education, gender, income, and age as independent variables to the TPB model and found an insignificant relationship with pro-environmental behavior. However, Nguyen [67] discovered that gender, age, and education level had a statistically significant impact on environmental awareness and attitude toward recycling.

Similarly, Pakpour [64] found that age, years of education, and gender were significantly correlated with past and follow-up household waste behavior. Bandara [97] discovered that those from a higher-income group were more likely to engage in waste reduction and separation activities than those from a lower-income group. The literature presents several research gaps on the correlation between external influential factors and waste recycling behavior. Prior studies reported contradicting outcomes for demographic factors, as well as recycling intention and behavior, across various countries. Based on the previous discussion, gender, education, and income (three demographic variables) were considered in this study. This study determined if socio-demographic variables have an impact on consumers' ERB in Malaysia using the extended TPB. Figure 1 shows the relationships of the extended TPB variables with socio-demographic variables as multi-group moderating effects.

## 3. Methodology

### 3.1. Sample

G*Power version 3.1 was used to compute the minimum sample size based on statistical power in order to determine the sample size for this study [98]. G*Power was often used to graphically display the relationship between the relevant variables and can compute minimum sample size and thus improve effect size calculation. Generally, the required power for social science and behavioral studies is 0.8 or above [99]. Because the model contains five predictors, several parameters were used to determine the sample size: medium effect size (0.15), $\alpha$ err probability 0.05, and a power of 0.95. After calculation using G*Power, the total minimum sample size was 138. The data collection process took seven months (March–October 2020) via an online questionnaire created using Google Forms and sent to the respondents based on the purposive sampling method. In total, 167 questionnaires were returned, but only 159 were complete for further analysis (8 questionnaires were incomplete). According to Sekaran [100], sample sizes that are greater than 30 and less than 500 are applicable for many studies. Meanwhile, Malhotra [101] indicated that a sample size of 150 is enough to conduct research. Hinkin [102] also indicated that the proper sample size should be between 1:4 and 1:10, depending on the item-to-response ratio. There were 35 items in this study; hence, the sample size should range from 140 to 350. As a result, we can safely conclude that a sample size of 159 was sufficient for this study and had sufficient statistical power. As such, the response rate was 95.2% for this study.

### 3.2. Research Instrument and Measurement

There are three techniques or methods for evaluating studies, including qualitative methods, quantitative methods, and mixed methods [103]. However, this study used the quantitative approach for data collection and analysis of the study. This is because this method can provide a valid demonstration and accuracy of the variables. This method is suitable to investigate the factors of intention and behavior toward e-waste recycling by testing the hypotheses that have been developed by using the extended TPB. Apart from that, it could provide indications of the individual's behavior in a large population with the same questions and can be beneficial to prove the validity of the study by using the results from the analysis. The quantitative method used in this study was survey research, which encompasses constructing a questionnaire to obtain primary data from the target respondents. The success of questionnaires is based on the fact that they are reasonably convenient to design, incredibly flexible, and highly competent in collecting a vast amount of data efficiently in a medium that is quickly processed [104]. The questionnaire was constructed using the Google Forms software application by Google. The study variables were adapted from previous studies that fit the context of this present research work. The questionnaire was composed of two parts: Parts A and B. Part A gathered the socio-demographic profile of the respondents, including gender, age, marital status, education, occupation, and income (see Table 1). Next, Part B was adopted and adapted from previous studies (see [23,45,70,86]) and consisted of attitude, SN, PBC, AEC, PC, MO, ERI, and ERB,

using a five-point Likert scale to measure the respondents' level of agreement with the items. The description of the constructs is provided in Table A1 (see Appendix A). After the stage of data collection, the data were screened, processed, and analyzed by using the Statistical Package for the Social Sciences (SPSS). Lastly, this study adopted PLS-SEM and utilized Smart PLS 3.2.9 software to conduct the statistical analysis for the proposed model. Moreover, the reflective measurement and structure model were used to test the reliability and validity of the study.

**Table 1.** Demographic profile of respondents.

| Demographic | Variables | Frequency (F) | Percentage (%) | Cumulative Percentage (%) |
|---|---|---|---|---|
| Gender | Male | 69 | 43.4 | 43.4 |
| | Female | 90 | 56.6 | 100.0 |
| Age | 18–30 | 71 | 44.7 | 44.7 |
| | 31–40 | 48 | 30.2 | 74.8 |
| | 41–50 | 21 | 13.2 | 88.1 |
| | 51–60 | 5 | 3.1 | 91.2 |
| | 61 and above | 14 | 8.8 | 100.0 |
| Marital | Single | 65 | 40.9 | 40.9 |
| | Married | 94 | 59.1 | 100.0 |
| Education | SPM | 14 | 8.8 | 8.8 |
| | Diploma | 43 | 27.0 | 35.8 |
| | Bachelor's degree | 69 | 43.4 | 79.2 |
| | Postgraduate | 31 | 19.5 | 98.7 |
| | Others | 2 | 1.3 | 100.0 |
| Occupation | Public Sector | 26 | 16.4 | 16.4 |
| | Private Sector | 60 | 37.7 | 54.1 |
| | Self-employed | 23 | 14.5 | 68.6 |
| | Student | 19 | 11.9 | 80.5 |
| | Pension | 6 | 3.8 | 84.3 |
| | Housewife | 13 | 8.2 | 92.5 |
| | Others | 12 | 7.5 | 100.0 |
| Income | MYR 2000 and below | 57 | 35.8 | 35.8 |
| | MYR 2001–MYR 3000 | 21 | 13.2 | 49.1 |
| | MYR 3001–MYR 4000 | 27 | 17.0 | 66.0 |
| | MYR 4001–MYR 5000 | 17 | 10.7 | 76.7 |
| | MYR 5001–MYR 6000 | 13 | 8.2 | 84.9 |
| | MYR 6001 and above | 24 | 15.1 | 100.0 |

## 4. Results

### 4.1. Data Screening

Skewness and kurtosis were conducted as an early test for multivariate normality [105] to verify if the study data are not too far from the normal distribution. The results reveal that the values of skewness and kurtosis ranged from −1.120 to 0.261 and −0.904 to 1.179, respectively. The skewness and kurtosis values for all the variables reflect an acceptable and normal distribution, signifying normality. Next, common method variance (CMV) was performed through Harman's single-factor test to avoid common method bias. The un-rotated factor analysis showed that the first factor accounted for only 38.36% of the variance. Hence, common method bias was not a serious threat to this study as the value was below 50% of the threshold value of variance [106].

### 4.2. Measurement Model

In the assessment of model structures in this study, the two-step process was executed. First, the outer model (measurement model) was assessed, and this was followed by the assessment of the inner model (structural model), which were developed using partial

least squares structural equation modeling (PLS-SEM) for data analysis. To assess the measurement model, consistency reliability, convergent validity, and discriminant validity were determined. Next, the assessment of the structural model was supported by the variance explanation of endogenous constructs, coefficient of determination, and predictive relevance. According to Hair [107], the composite reliability (CR) of each construct should exceed 0.70 to reflect good reliability. The CR values listed in Table 1 for attitude, SN, PBC, AEC, PC, MO, ERI, and ERB are 0.913, 0.938, 0.921, 0.916, 0.890, 0.897, 0.877, and 0.832, respectively. These values indicate that each variable recorded a satisfactory and high level of internal consistency. Hence, CR can be applied to measure internal consistency reliability precisely.

Next, the acceptable outer indicator loading should exceed 0.70. Table 2 and Figure 2 show that all indicator loadings are above 0.70, except for Attitude1 (0.648) and MO1 (0.655). However, these indicators were maintained because outer loading that ranges from 0.4 to 0.7 is acceptable if CR ≥ 0.7 or AVE ≥ 0.5 [99,108]. Referring to Table 2, the measurement items in this study reached a satisfactory level of reliability, and each item represents common constructs. An AVE value of 0.50 or higher signifies a sufficient degree of convergent validity, which means that the latent variable explains at least half of its indicators' variance [107]. In Table 2, the AVE value of each latent variable of this study is above 0.50. This confirms that the measurement scale deployed in this study correctly represents the constructs.

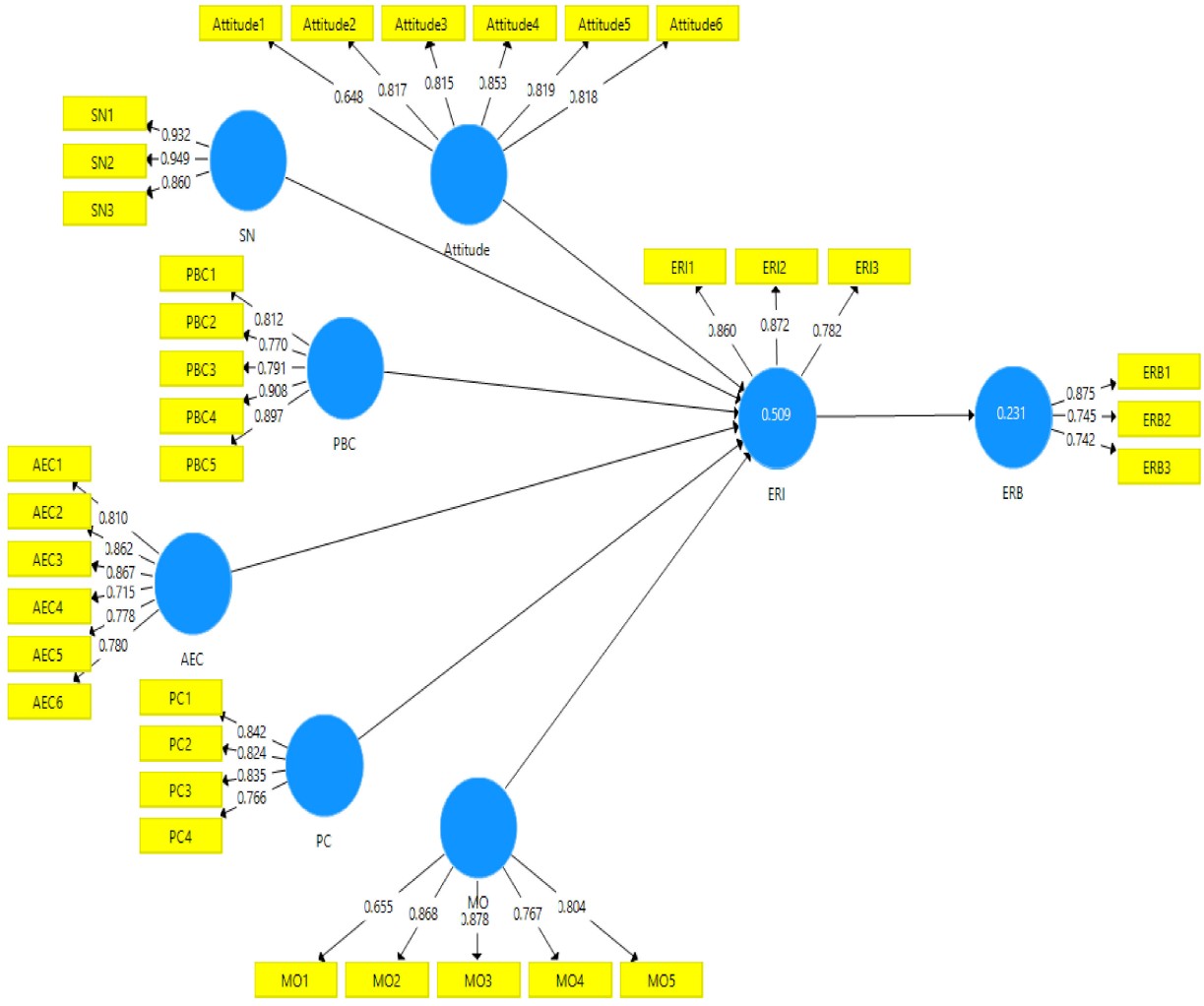

**Figure 2.** The measurement model.

**Table 2.** Summary of the measurement model.

| Construct | Indicator | Indicator Loading | Cronbach's Alpha | Composite Reliability | Average Variance Extracted (AVE) |
|---|---|---|---|---|---|
| Attitude | Attitude1 | 0.648 | 0.885 | 0.913 | 0.636 |
| | Attitude2 | 0.817 | | | |
| | Attitude3 | 0.815 | | | |
| | Atitude4 | 0.853 | | | |
| | Attitude5 | 0.819 | | | |
| | Attitude6 | 0.818 | | | |
| Subjective Norm (SN) | SN1 | 0.932 | 0.901 | 0.938 | 0.836 |
| | SN2 | 0.949 | | | |
| | SN3 | 0.860 | | | |
| Perceived Behavioral Control (PBC) | PBC1 | 0.812 | 0.892 | 0.921 | 0.701 |
| | PBC2 | 0.770 | | | |
| | PBC3 | 0.791 | | | |
| | PBC4 | 0.908 | | | |
| | PBC5 | 0.897 | | | |
| Awareness of Environmental Consequences (AEC) | AEC1 | 0.810 | 0.889 | 0.916 | 0.646 |
| | AEC2 | 0.862 | | | |
| | AEC3 | 0.867 | | | |
| | AEC4 | 0.715 | | | |
| | AEC5 | 0.778 | | | |
| | AEC6 | 0.780 | | | |
| Perceived Convenience (PC) | PC1 | 0.842 | 0.835 | 0.890 | 0.668 |
| | PC2 | 0.824 | | | |
| | PC3 | 0.835 | | | |
| | PC4 | 0.766 | | | |
| Moral Obligation (MO) | MO1 | 0.655 | 0.854 | 0.897 | 0.637 |
| | MO2 | 0.868 | | | |
| | MO3 | 0.878 | | | |
| | MO4 | 0.767 | | | |
| | MO5 | 0.804 | | | |
| E-waste Recycling Intention (ERI) | ERI1 | 0.860 | 0.790 | 0.877 | 0.704 |
| | ERI2 | 0.872 | | | |
| | ERI3 | 0.782 | | | |
| E-waste Recycling Behavior (ERB) | ERB1 | 0.875 | 0.707 | 0.832 | 0.624 |
| | ERB2 | 0.745 | | | |
| | ERB3 | 0.742 | | | |

*4.3. Discriminant Validity*

Discriminant validity is measured by evaluating the Fornell–Larcker criterion and HTMT ratio. The Fornell–Larcker criterion denotes the concept that a construct shares more variance with its related indicators than with other constructs. The square root of each AVE construct should exceed its highest correlation with other constructs. The assessment of discriminant validity using Fornell and Larcker [109] and HTMT is shown in Tables 3 and 4, respectively. The analysis of the Fornell–Larcker criterion tabulated in Table 3 shows the square root values of AEC (0.804), attitude (0.798), ERB (0.790), ERI (0.839), MO (0.798), PBC (0.838), PC (0.818), and SN (0.914), hence, it showed that every indicator's loadings were higher than the cross loadings of other construct. Thus, the discriminant validity in this study was established. Meanwhile Table 4 lists the discriminant validity using the HTMT criterion. For the HTMT criterion, all constructs should have values below 0.90. In Table 4, the discriminant validity shows that all construct values are below 0.90 [110].

**Table 3.** Fornell–Larcker criterion for discriminant validity.

|  | AEC | Attitude | ERB | ERI | MO | PBC | PC | SN |
|---|---|---|---|---|---|---|---|---|
| AEC | **0.804** | | | | | | | |
| Attitude | 0.666 | **0.798** | | | | | | |
| ERB | 0.344 | 0.270 | **0.790** | | | | | |
| ERI | 0.465 | 0.469 | 0.481 | **0.839** | | | | |
| MO | 0.659 | 0.630 | 0.377 | 0.674 | **0.798** | | | |
| PBC | 0.440 | 0.389 | 0.486 | 0.472 | 0.539 | **0.838** | | |
| PC | 0.457 | 0.356 | 0.513 | 0.502 | 0.469 | 0.751 | **0.818** | |
| SN | 0.365 | 0.378 | 0.428 | 0.487 | 0.487 | 0.674 | 0.663 | **0.914** |

Note: SN = Subjective Norm, PBC = Perceived Behavioral Control, AEC = Awareness of Environmental Consequences, PC = Perceived Convenience, MO = Moral Obligation, ERI = E-waste Recycling Intention, ERB = E-waste Recycling Behavior.

**Table 4.** Discriminant validity assessment (HTMT).

|  | AEC | Attitude | ERB_ | ERI_ | MO | PBC | PC | SN |
|---|---|---|---|---|---|---|---|---|
| AEC | | | | | | | | |
| ATTITUDE | 0.745 | | | | | | | |
| ERB_ | 0.412 | 0.319 | | | | | | |
| ERI_ | 0.533 | 0.529 | 0.616 | | | | | |
| MO | 0.754 | 0.71 | 0.47 | 0.799 | | | | |
| PBC | 0.502 | 0.428 | 0.605 | 0.562 | 0.612 | | | |
| PC | 0.533 | 0.401 | 0.665 | 0.62 | 0.546 | 0.875 | | |
| SN | 0.41 | 0.422 | 0.533 | 0.58 | 0.556 | 0.759 | 0.766 | |

Note: SN = Subjective Norm, PBC = Perceived Behavioral Control, AEC = Awareness of Environmental Consequences, PC = Perceived Convenience, MO = Moral Obligation, ERI = E-waste Recycling Intention, ERB = E-waste Recycling Behavior.

### *4.4. Assessment of Structural Model*

After verifying the validity and the reliability of the study constructs, the structural model was examined. According to Hair [99], a collinearity assessment should be performed to prevent a biased path coefficient. To ensure the absence of multicollinearity issues, the tolerance should exceed 0.20 or the VIF must be less than 5 [107]. By analyzing the inner VIF values in SmartPLS, the following were recorded: Attitude_ERI (VIF = 2.058), SN_ERI (VIF = 2.137), PBC_ERI (VIF = 2.802), AEC_ERI (VIF = 2.293), PC_ERI (VIF = 2.677), MO_ERI (VIF = 2.313), and ERI_ERB (VIF = 1.000). The outcomes reveal that the largest VIF of constructs was 2.802, which is below the threshold value and proves the absence of a multicollinearity issue.

The second stage involved to assess the inner model is to identify the importance and the relevance of the relationships in the structural model by analyzing the beta values (β) of the path coefficients. Referring to Figure 3, five positive path coefficient values and two negative coefficient values are noted. The highest path coefficient denotes the path of MO → ERI (0.543), followed by ERI → ERB (0.481), PC → ERI (0.221), SN → ERI (0.115), PBC → ERI (−0.067) (negative direction), attitude → ERI (0.063), and finally, AEC → ERI (−0.049) (negative direction). Based on the bootstrapping in PLS-SEM, a *t*-value was obtained to assess significant relationships of the path coefficients followed by hypothesis testing.

The coefficient of determination ($R^2$) measures the predictive accuracy of a model [99]. Larger $R^2$ values indicate higher predictive accuracy [99]. Figure 2 shows that ERI has an $R^2$ value of 0.509, while ERB on the implementation of ERI has an $R^2$ value of 0.231, which are acceptable and sufficient to prove that the model has moderate predictive accuracy.

Next, the blindfolding procedure was executed for the assessment of $Q^2$ of the path model [99]. A smaller variance between the predicted and the original values depicts a greater $Q^2$ and predictive accuracy of the model. As the $Q^2$ values of ERI ($Q^2$ = 0.323) and ERB on implementation of ERI ($Q^2$ = 0.133) exceeded 0, they reflect that the model has predictive relevance power.

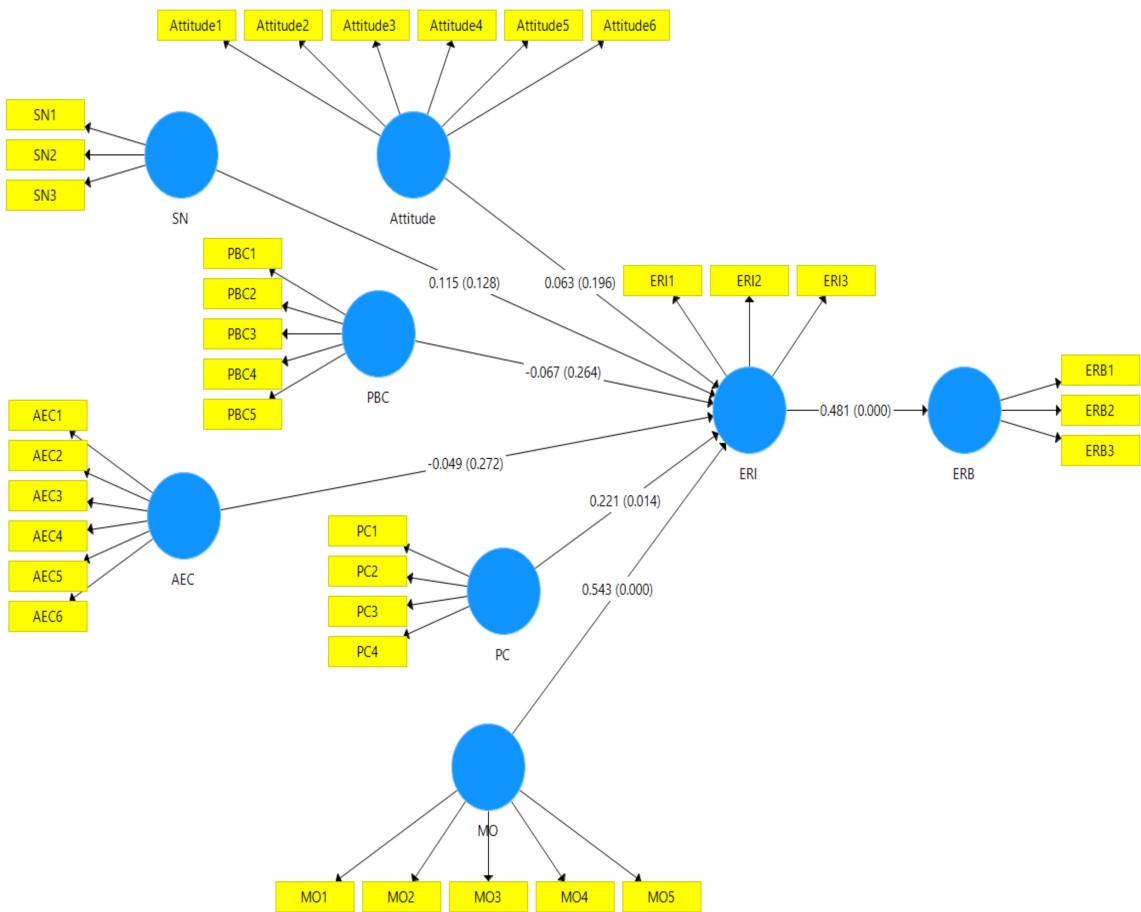

**Figure 3.** The structural model.

In order to examine the seven proposed hypotheses, the PLS bootstrapping technique was performed to assess the significant relationship of each path coefficient. *T*-values and *p*-values were used to determine if the hypotheses were accepted or rejected. To perform the PLS bootstrapping, the following settings were applied: (a) 5000 samples, (b) 159 cases, and (c) a one-tailed type test with a 95% significance level ($p < 0.05$). Referring to Table 5, Hypothesis 1 shows that attitude positively influences ERI, while Hypothesis 2 presents that SN positively influences ERI. Although the path coefficients for attitude → ERI (0.063) and SN → ERI (0.115) are positive, these two hypotheses are not supported as the values of *t*-statistics and *p*-value for attitude → ERI and SN → ERI are *T*-values = 0.854 and 1.135, <1.96, and *p*-values = 0.196 and 0.128, >0.05. Meanwhile, Hypothesis 3 denotes that PBC negatively influences ERI, and Hypothesis 4 presents that AEC negatively influences ERI. The path coefficients for PBC → ERI (−0.067) and AEC → ERI (−0.049) are negative, whereas the values of *t*-statistics and *p*-value for PBC → ERI and AEC → ERI are *t*-value = 0.631 and 0.606, <1.96, and *p*-value = 0.264 and 0.272, >0.05. Therefore, Hypotheses 3 and 4 are not supported either. Moving on, Hypothesis 5 shows that PC positively influences ERI, Hypothesis 6 depicts that MO positively influences ERI, and Hypothesis 7 presents that ERI positively influences ERB—these three hypotheses are supported. The path coefficients for PC → ERI (0.221), MO → ERI (0.543), and ERI → ERB (0.481) are positive. The values of *t*-statistics and *p*-value for PC → ERI, MO → ERI, and ERI → ERB are *T*-value = 2.203, 6.931, and 7.413, >1.96, and *p*-value = 0.014, 0.000, and 0.000, <0.05. Based on Table 5, the results of the tested hypotheses show that only Hypotheses 5, 6, and 7 are supported. However, Hypotheses 1, 2, 3, and 4 are not supported, as illustrated in Figure 3.

**Table 5.** Results of hypothesis testing.

| Hypothesis | Path Coefficient | Standard Error | *t*-Statistics | *p*-Value | Result |
|---|---|---|---|---|---|
| H1: Attitude → ERI | 0.063 | 0.074 | 0.854 | 0.196 | Not Supported |
| H2: SN → ERI | 0.115 | 0.121 | 1.135 | 0.128 | Not Supported |
| H3: PBC → ERI | −0.067 | 0.106 | 0.631 | 0.264 | Not Supported |
| H4: AEC → ERI | −0.049 | 0.082 | 0.606 | 0.272 | Not Supported |
| H5: PC → ERI | 0.221 | 0.100 | 2.203 | 0.014 | Supported |
| H6: MO → ERI | 0.543 | 0.078 | 6.931 | 0.000 | Supported |
| H7: ERI → ERB | 0.481 | 0.065 | 7.413 | 0.000 | Supported |

Note: SN = Subjective Norm, PBC = Perceived Behavioral Control, AEC = Awareness of Environmental Consequences, PC = Perceived Convenience, MO = Moral Obligation, ERI = E-waste Recycling Intention, ERB = E-waste Recycling Behavior. (The path is significant at *p*-value = 0.05).

### 4.5. Multi-Group Analysis (Age, Education, and Income)

Multi-group analysis (MGA), as presented in Table 6, shows the impacts of socio-demographic factors, namely gender, education, and income. In this study, gender is a categorical variable, while education and income are split into two groups. The education factor is split into a high education group (postgraduate) and a low/middle education group (undergraduate or diploma). As for the income factor, it is split into a low-income group (MYR 2000 and below) and a high-income group (MYR 6000 and above). Henseler [111] extended this method by initiating the PLS-MGA technique (MGA), after recognizing the significant differences within groups when lower than or equal to 0.05 or above 0.95. The percentile bootstrapping method was used to assess the differences between groups in this study. As a result, a substantial inter-group variance was noted with an error margin of 5% when the *p*-value exceeded 95% and was less than 5%.

**Table 6.** The MGA analysis.

| | H1 | H2 | H3 | H4 | H5 | H6 | H7 |
|---|---|---|---|---|---|---|---|
| **Gender** | | | | | | | |
| Female | 0.161 | 0.132 | −0.217 | −0.039 | 0.228 | 0.606 | 0.518 |
| Male | 0.016 | 0.178 | 0.077 | −0.053 | 0.178 | 0.411 | 0.451 |
| Diff | 0.145 | 0.046 | 0.294 | 0.014 | 0.049 | 0.195 | 0.067 |
| PLS MGA Value | 0.159 | 0.591 | 0.915 | 0.512 | 0.417 | 0.179 | 0.285 |
| **Education** | | | | | | | |
| High | −0.137 | −0.064 | 0.243 | 0.050 | −0.205 | 0.717 | 0.562 |
| Low/Middle | −0.182 | 0.393 | −0.252 | −0.011 | 0.422 | 0.499 | 0.647 |
| Diff | 0.044 | 0.456 | 0.495 | 0.062 | 0.627 | 0.219 | 0.086 |
| PLS MGA Value | 0.425 | 0.917 | 0.117 | 0.435 | **0.982** | 0.275 | 0.695 |
| **Income** | | | | | | | |
| High | 0.367 | 0.497 | 0.098 | 0.271 | −0.506 | 0.117 | 0.752 |
| Low | 0.011 | 0.022 | −0.078 | −0.022 | 0.279 | 0.649 | 0.578 |
| Diff | 0.356 | 0.474 | 0.176 | 0.292 | 0.784 | 0.532 | 0.174 |
| PLS MGA Value | 0.131 | 0.130 | 0.333 | 0.184 | 0.935 | 0.898 | 0.075 |

Note: Bold font: PLS-MGA *p*-values below 5% and above 95% indicate significant values. Diff = Path Coefficient Differences.

After running the PLS-MGA, the *p*-value shows that only education, H5 (*p* = 0.982), displayed a significant effect—the relationship between PC and ERI is stronger for the low/middle education group than for the high education group. However, gender and income level exhibited insignificant effects on ERI.

## 5. Discussion and Conclusions

The continuous production of e-waste has led to a massive threat to the living environment, thus adversely affecting the sustainable economic growth of many countries [112]. Therefore, an effective regulatory system for the proper disposal of e-waste and an efficient mechanism of a proper take-back system from the public and industries should be implemented. Simply put, recycling by reverse logistics should be promoted to reduce environmental issues, while concurrently generating economic advantages for organizations. Unfortunately, studies that link reverse logistics systems to consumer behavior are in scarcity [25]. Notably, the reverse logistics process cannot be operated without consumer participation because they are the initial connection in the entire supply chain [26]. In order to bridge this gap, this present study assessed factors that influence ERI and ERB in Malaysia through the lens of an extended TPB model by adding three variables, namely AEC, PC, and MO.

First, the finding revealed that the relationship between attitude and ERI (H1) is insignificant. The result contradicts that reported in past studies (see [23,24,31,45,57,67–69]), which depicted that attitude toward e-waste recycling positively determines the intention to perform recycling. According to Greaves [57], recycling attitudes can be formed by having a favorable assessment of recycling, for instance, a person should assume that waste disposal is responsible, conscientious, and convenient to perform. Moreover, Echegaray and Hansstein [63] stated that attitude recycling is motivated by the belief that recycling is beneficial for both the environment and human health. Based on the results, the study respondents agreed that recycling e-waste is responsible, pleasant, beneficial, and sensible. However, this attitude factor did not encourage them to actually perform the recycling behavior. This happened because Malaysians still have limited knowledge of e-waste and its consequences on environmental and human health. Previous studies have explained that some people refuse to make fair use of their positive attitude toward recycling intentions and behavior [113]. In Malaysia, consumers do not perform recycling activities because they believe recycling is a voluntary activity with no benefit or incentive [114]. That is why Chan and Bishop [35] introduced a new TPB framework by replacing attitude elements with moral norms to study recycling intention and behavior because previous researchers [35,71,74] have identified that internalized norms such as moral obligation can be predictors of intention, attitude, or a replacement for attitude in the research of general conservation behavior that includes components about recycling.

Next, the association between SN and ERI (H2) was insignificantly negative in this study. Prior studies (see [72,76,115]) are in line with this outcome, mainly because SN is a statistically insignificant predictor of recycling intention. This insignificant result of the SN–ERI link in this study denotes that Malaysians, especially the young generation, do not feel pressure from individuals that they consider important to them (e.g., friends, colleagues, mass media, and society) to perform recycling activities [76,115]. One possible explanation for the insignificant effect of SN is that the society in Malaysia barely performs any waste segregation or recycling activities [116]. Moving on, Hypothesis 3 (H3) indicates an insignificant PBC–ERI relationship. The finding contradicts that of past studies, which reported a positive impact of PBC on one's recycling intention (see [24,64,68,72,92]). This insignificant result is ascribed to appropriate recycling channels which are rather difficult to find in Malaysia. Moreover, the study respondents did not perform e-waste recycling because they were unsure which e-waste items could be recycled. Nonetheless, this finding is in line with that found by Echegaray and Hansstein [63]; the PBC emerged as the most negligible influential factor when compared to other factors. This notion is attributed to the inadequate supply of solid garbage collection facilities, as well as inadequate information and resources about e-waste recycling. Hence, it is rather difficult for them to decide if recycling is practical to perform as part of their ordinary activities [23].

Next, the result shows that the relationship between AEC and ERI (H4) is not supported in this study. On the contrary, Echegaray and Hansstein [63] reported that awareness of environmental issues and consideration for the state of the environment forecast a posi-

tive disposition toward recycling behavior. Nonetheless, consumers become unaware of the consequences of not recycling e-waste because it is not seen as a lucrative or environment-friendly practice [23]. Afroz [77] revealed that although most Malaysian consumers were aware of the adverse effect of electronic goods on environmental and human health, only 2–3% of them were interested in recycling those goods. Even though most consumers believe that e-waste recycling is vital, they are still unsure if e-waste is a serious threat to the environment [45]. The insignificant result retrieved for the AEC–ERI link in this study is ascribed to the awareness and concern for the environment and recycling not being nurtured from an early age in Malaysia. Hence, the education system at both primary and secondary levels in Malaysia should foster and encourage recycling behavior [114].

Next, this study recorded a significantly positive link between PC and ERI (H5). This result is in agreement with prior studies (see [2,27,79,117]) that PC is a significant factor in motivating consumers' intention to recycle. This signifies that most Malaysian consumers feel that convenience, particularly time, space, and distance, will motivate them to recycle their e-waste. Notably, e-waste recycling may be increased with adequate time to organize e-waste, effort to clear personal information on electronic goods, and sufficient storage space. The respondents agreed that recycling e-waste does not take much time and room. The respondents agreed that implementing more recycling bins at strategic points can make recycling more convenient, as well as improve their participation in recycling activity [118].

Apparently, MO emerged as the strongest predictor of ERI (H6) in this study. Similarly, Razali [37] and Saphores [79] reported that MO is the most influential determinant of their extended TPB models. Turning to this present study, internal motivation (i.e., MO) can increase consumers' ERI. This result justifies the role of psychological variables, such as intrinsic motivation, in specific behavior, including pro-environmental behavior [37]. Tonglet [70] claimed that someone who perceives it is necessary to recycle or vice versa is likely to include personal norms in the decision-making process. This is because the encouragement to perform recycling behavior may be low if there is no incentive, and e-waste return may be costly if customers need to go to the disposal center to deposit their e-waste. This return action concentrates on morals instead of rationale, stemming from MO [31]. Chan and Bishop [35] initiated a new TPB framework by replacing the attitude element with moral norm to assess recycling intention and behavior. This is because past studies (see [35,81,119]) have identified that internalized norms, such as MO, may serve as predictors of intention and attitude or as a replacement for attitude in studies related to general conservation behavior that includes recycling components. This finding proves that intrinsic motivation is more influential in motivating consumers to recycle their e-waste than receiving extrinsic incentives.

A significantly positive relationship was observed between ERI and ERB (H7) in this study, which is in agreement with previous studies (see [35,64,68,84,88]). In a similar vein, Shaharudin [76] showed that Malaysians intending to discard their e-waste would inevitably affect their execution of ERB, especially when their electronic goods are damaged and have outdated features or models. Malaysians, especially the young generation, prefer adopting the appropriate method to manage their e-waste so that they too can conserve and preserve the environment [76]. Ho [86] and Echegaray and Hansstein [63] claimed that intentions have strong connections with behavior. Hence, ERB can be increased when the respondents have the intention to recycle e-waste, drop off their e-waste at an authorized facility, or return their e-waste to the retailer or manufacturer [45].

Lastly, this study investigated the effects of socio-demographic variables as moderating factors on recycling behavior under varying socio-demographic backgrounds (gender, education, and income) by employing the MGA-PLS method. First, the MGA for gender was divided into female and male. Similar to that reported by do Valle [94] and Botetzagias et al. [72], gender had no statistically significant relationship with ERI. The result proves that there are no significant differences between males and females in recycling behaviors in Malaysia. Second, the MGA-PLS was executed to determine the effect of the high education (postgraduate level) and low/middle education (undergraduate or diploma level) groups.

As a result, only PC and ERI passed the significance test. The lower/middle education group displayed a more significant effect on ERI or ERB than the higher education group. The results suggest that people with a lower/middle level of education concurred that convenience, particularly in terms of time, space, and distance, will encourage them to recycle their electronic waste. Finally, the findings show the multi-group effect regarding the low-income group (MYR 2000 and below) and the high-income group (MYR 6000 and above). Apparently, income level had no impact on ERI and ERB. Similarly, Botetzagias [72], Nguyen [67], and Wang [92] found that income level was insignificant when determining the intention of consumers to recycle e-waste. This proved that the low or poor performance of Malaysian consumers in recycling e-waste is not primarily caused by their level of income. Overall, socio-demographic variables were discovered to be statistically non-significant predictors of ERI and ERB, except for education.

### 5.1. Implications

This study provides practical and theoretical implications that may benefit academicians, the government, and industrial players. From the practical stance, the study outcomes are crucial because they offer several practical implications for the government and the industry or manufacturers, especially in Malaysia, in terms of implementing effective e-waste recycling initiatives to promote social sustainability. Notably, MO emerged as the most significant factor in influencing ERI and ERB among consumers. Hence, the government should be able to strengthen the regulations and implementations of e-waste recycling activities that place more focus on consumers' intrinsic motivation, while concurrently devising an effective method to foster ERB as MO among consumers in Malaysia. The government can promote e-waste recycling campaigns or programs that illustrate the significance of ERB and how it is viewed as a necessary behavior among consumers. Consumers should be advised about how ERB can be practiced on a daily basis and view it as their part of MO. According to Razali [37], extrinsic motivation may serve as the pressure that triggers the behavior, whereas intrinsic motivation encourages people to continue and maintain practicing and performing the behavior in the future. Thus, intrinsic motivation should be enhanced and strengthened instead of concentrating on the provision of extrinsic incentives.

The results reveal that PC displayed a positive influence on ERI. Therefore, undeniably, making recycling more convenient enhances recycling intentions among consumers, thus increasing the recycling rate. This study proved that the impact of the logistic structure of waste disposal, including the proximity of recycling bins and the allocation of curbside collection, is vital to achieve greater public involvement in recycling. The involvement of all stakeholders is integral, including consumers, non-government agencies, collectors, retailers, and recycling facilities. Manufacturers and retailers must collaborate with third-party logistics companies to reduce the volume of e-waste via reverse logistics to effectively manage discarded and outdated electronic goods. Effective recycling driven by take-back and collection initiatives generates the reverse logistics strategy of businesses [45].

Additionally, the implementation of an efficient e-waste management method should be strengthened and enhanced by deploying relevant legislation and regulations. In Malaysia, e-waste is listed as scheduled waste under the Environmental Quality (Scheduled Wastes) Regulations 2005. It specifies that no individual shall be permitted to dispose of any e-waste in a landfill, e-waste should be recycled and recovered at authorized recovery facilities, and the disposal process should take place only at licensed recycling facilities in an environment-friendly manner. The DOE has drafted a new regulation, known as the Environmental Quality (Household Scheduled Waste) Regulation, which is under review by the Attorney General's Chambers of Malaysia (AGC Malaysia). Specifically, the government shoulders the responsibility to develop rules and regulations, monitor the operation of recovery facilities, and impose a penalty on unauthorized recovery facilities. Simultaneously, consumers have the responsibility to gather and dispose of their e-waste at authorized recycling facilities.

The government should also implement more educational campaigns regarding e-waste recycling. Such campaigns can increase awareness and attitude among Malaysians about the importance of recycling to converse our natural resources, as well as minimize the use of landfills and greenhouse gas emissions. When consumers realize how beneficial it is to recycle their e-waste, it will encourage them to recycle their e-waste in the future and promote e-waste recycling practices to others. In light of the insignificant link between SN and ERI, social media and marketing strategies in Malaysia should play a significant role and be better planned in order to motivate more individuals to participate in recycling activities. As for the low PBC score, educational and engagement actions should be taken to educate Malaysians on the appropriate methods to recycle and reuse their e-waste, as well as the types of e-waste elements that can be recycled. Another effective method that may convince consumers to engage in formal e-waste recycling activity is providing adequate resources for recycling e-waste. In fact, the insignificant PBC–ERI link observed in this study revealed that recycling behavior can be disrupted by the inconvenience of recycling. Hence, the necessity to ensure the availability and accessibility of e-waste disposal facilities should be of top concern.

From the theoretical perspective, this study contributes to the body of knowledge by extending the TPB framework in terms of ERB among consumers in Malaysia. The TPB model was extended by embedding attitude, SN, and PBC. However, some setbacks were noted when this model was applied in the context of pro-environmental behavior in past studies. For instance, pro-environmental behavior that involved skills, resources, accessibility, and control was poorly predicted by the TPB model. Therefore, the extended TPB model was proposed in this study to fill the gap through the inclusion of AEC, PC, and MO. With that, new findings were reported in this study pertaining to attitude, SN, PBC, AEC, PC, and MO with ERI, as well as ERI with ERB, among consumers in Malaysia.

In this study, MO emerged as the most influential factor in enhancing the intrinsic motivation of consumers to participate in ERI and ERB. This proves that intrinsic motivation is essential when adopting specific behavior, such as recycling e-waste among Malaysian consumers. Consumers display a greater sense of MO because they will feel right or wrong about what they should perform or should not. For instance, ERI increases if they feel guilty when they could not perform e-waste recycling activity and it is against their principles not to recycle e-waste. This study revealed how more compelling intrinsic motivation is than extrinsic rewards. MO is identified as positive behavior that should be fostered and embraced by all consumers in Malaysia.

The second highest score denotes the relationship between ERI and ERB. This proves that intentions to recycle have strong correlations with recycling behavior. The respondents agreed that if they have the intention to drop off or return their e-waste at a nearby recycling station or retailer, they would probably perform the intended recycling behavior in the future.

Next, PC displayed a significantly positive link to ERI. This means convenience can influence ERI among consumers in Malaysia. Apparently, the respondents agreed that when recycling is more convenient, their recycling intentions will be enhanced, and this can effectively increase the recycling rate. This study discovered that the conditions of the logistics chain of waste disposal, such as the physical closeness of bins or the availability of curbside waste, can affect the participation of the public in recycling activities. Performing e-waste recycling is eased when they have adequate space to store recycled e-waste at home and have sufficient time to recycle their e-waste. These should intensify their recycling intentions.

Notably, the relationships of ERI with attitude, SN, PBC, and AEC were insignificant in the context of Malaysian consumers. This calls for further investigations, mainly because past studies (see [17,30,37,87,120]) found that attitude, SN, PBC, and AEC had a positive impact on behavioral intention.

### 5.2. Future Research Endeavor and Limitations

This study had several limitations. As the concepts of e-waste and e-waste recycling are still at their early stages in Malaysia, most of the consumers are still unfamiliar with e-waste. Moreover, this study used a limited sample size, and the results show that the relationship of ERI with attitude, SN, PBC, and AEC emerged as insignificant in the study, although in previous studies, the independent variables were strong predictors of recycling behavioral intention. For instance, Wan [52] and Tonglet [70] reported that attitude is the most significant determinant of recycling intention. Therefore, it is recommended that future empirical studies introduce and construct attitude from a different conceptual view, such as conceptualizing attitude as the willingness [121] and attitude of eagerness [122] toward pro-environmental studies such as e-waste recycling behavior. Meanwhile, other studies (see [23,30,123,124]) highlighted that SN is a crucial influencer of waste recycling intention. In addition, Pakpour [64] indicated that PBC does not just predict behavioral intention, but it can also predict behavior and intention. On top of that, many studies have proven that AEC can positively predict recycling intentions (see [32,45,52,63]). Thus, future researchers may expand their sample size to test the extended TPB model.

Although the sample size ($n = 159$) deployed in this study meets the required minimum sample size of G*Power ($n = 138$), it failed to represent the whole population in Malaysia aged 18 years and above with the purchasing power of electrical and electronic products, as well as with the intention to recycle electrical and electronic goods. Therefore, in order to explore the changing behavior of consumers, future studies should focus on a specific group. For example, the specific group can be low-income consumers, high-income consumers, youth consumers, or even university consumers. These specific groups could establish a more comprehensive explanation of why consumers refuse to participate in ERB. Past studies (see [25,45,125]) have assessed university students as their target respondents to study recycling intention and behavior. Meanwhile, Shaharudin [52] studied youth and Kumar looked into young adults to determine ERI and ERB.

Next, the survey questionnaire method was executed online using Google Forms in this study. This survey method led to some shortcomings in this study. Despite being a free online tool that allows researchers to create surveys within a short time and gather data easily, Google Forms involves a number of disadvantages including respondents facing poor Internet connection to access Google Forms. Hence, future researchers should distribute the questionnaire forms manually to the target respondents. In addition, researchers may also provide an explanation to the respondents to ensure that the respondents completely understand the questions before providing responses.

In this study, ERB on ERI implementation scored an $R^2$ value of 0.231 (rather weak), implying that the variable displayed poor fit in this model. Hence, future researchers may use or add additional variables as the extended determinants in the TPB to enhance the predictive accuracy of the model.

**Author Contributions:** Conceptualization, N.S.M. and A.C.T.; methodology, N.S.M.; software, N.S.M.; validation, N.S.M., A.C.T. and H.T.H.; formal analysis, N.S.M.; investigation, N.S.M.; resources, N.S.M.; data curation, N.S.M. and A.C.T.; writing—original draft preparation, N.S.M.; writing—review and editing, A.C.T. and H.T.H.; visualization, A.C.T. and H.T.H.; supervision, A.C.T. and H.T.H.; project administration, N.S.M. and A.C.T.; funding acquisition, A.C.T. All authors have read and agreed to the published version of the manuscript.

**Funding:** This research received no external funding.

**Institutional Review Board Statement:** The study did not require separate ethics board approval, but it did adhere to the ethics guidelines of the authors' institutions.

**Informed Consent Statement:** Not applicable.

**Data Availability Statement:** Not applicable.

**Conflicts of Interest:** The authors declare no conflict of interest.

**Appendix A**

**Table A1.** Measurement items.

| Constructs/Items | Construct/Items Description | Source |
|---|---|---|
| | Attitude | |
| Attitude1 | E-waste recycling is pleasant | |
| Attitude2 | E-waste recycling is responsible | |
| Attitude3 | E-waste recycling is good | Kumar [23] |
| Attitude4 | E-waste recycling is beneficial | |
| Attitude5 | E-waste recycling is rewarding | |
| Attitude6 | E-waste recycling is sensible | |
| | Subjective norms | |
| SN1 | My friends expect me to recycle e-waste | |
| SN2 | My classmates or colleagues expect me to recycle e-waste | Kochan [45] |
| SN3 | The media influence me to recycle e-waste | |
| | Perceived behavioral control | |
| PBC1 | I know what items of e-waste can be recycled | |
| PBC2 | I have plenty of opportunities to recycle e-waste | |
| PBC3 | The local council provides satisfactory resources for recycling e-waste | Tonglet [70] |
| PBC4 | I know where to take my e-waste for recycling | |
| PBC5 | I know how to recycle my e-waste | |
| | Awareness of environmental consequences | |
| AEC1 | Recycling e-waste conserves energy | |
| AEC2 | Recycling e-waste preserves natural resources | |
| AEC3 | Recycling e-waste reduces pollution | Ho [86] |
| AEC4 | Recycling e-waste saves money | |
| AEC5 | Recycling e-waste reduces the use of landfills | |
| AEC6 | Recycling e-waste protects human health | |
| | Perceived convenience | |
| PC1 | There is enough space for me to keep my recycled e-waste at home | |
| PC2 | Recycling my e-waste is convenient | Ho [86] |
| PC3 | I have convenient access to a drop-off center for e-waste recycling | |
| PC4 | I have time to recycle my e-waste | |
| | Moral obligation | |
| MO1 | I feel I should not waste any electronic good if it could be re-used | |
| MO2 | It would be wrong of me not to recycle my e-waste | |
| MO3 | I would feel guilty if I did not recycle my e-waste | Tonglet [70] |
| MO4 | Not recycling my e-waste goes against my principles | |
| MO5 | Everybody should share the responsibility to recycle e-waste | |

**Table A1.** *Cont.*

| Constructs/Items | Construct/Items Description | Source |
|:---:|:---:|:---:|
| | E-waste recycling intention | |
| ERI1 | I intend to recycle e-waste regularly | |
| ERI2 | I intend to drop-off e-waste at a nearby recycling station | Kochan [45] |
| ERI3 | I intend to return e-waste to the retailer or the manufacturer | |
| | E-waste recycling behavior | |
| ERB1 | I donate e-waste | |
| ERB2 | I resell e-waste | Kochan [45] |
| ERB3 | I store e-waste | |

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
