# Peer review of "The Determinants of Consumers’ E-Waste Recycling Behavior through the Lens of Extended Theory of Planned Behavior"

_sustainability, doi:10.3390/su14159031_

Round 1
Reviewer 1 Report
The article titled The Determinants of Consumers’ E-Waste Recycling Behavior through the Lens of Extended Theory Planned Behavior uses well established theory to explore recycling behavior.
Overall the paper is of good quality. However improvements are mandatory.
1. The TPB has been well used and is known to explain behavior. Emphisize why the paper is original, and innovative.
2. The TPB has also been used to explore other behaviors in recent years such as OCB, knowledge sharing. And some new elements were introduced to the model, where attitude is conceptualized as willingness and eagerness. (you could also suggest that in future studies)
Some useful literature on the topic
Obrenovic, B., Jianguo, D., Tsoy, D., Obrenovic, S., Khan, M. A. S., & Anwar, F. (2020). The enjoyment of knowledge sharing: impact of altruism on tacit knowledge-sharing behavior. Frontiers in Psychology, 11, 1496.
Bakar, N.A., Rosbi, S., Hashim, H. & Arshad, N.C. (2021). Factors Influencing Students Intention to Choose Career of Halal Food Industry in Malaysia using Theory of Planned Behavior. International Journal of Management Science and Business Administration, 8(1), 50-67.
Obrenovic, B., Du, J., Godinić, D., & Tsoy, D. (2021). Personality trait of conscientiousness impact on tacit knowledge sharing: the mediating effect of eagerness and subjective norm. Journal of Knowledge Management.
3. The study sample is very small. Please elaborate.
4. The methods and data collection is not sufficiently described to be replicated. You should provide more information on it. Sampling is questionable.
5. Include descriptives in the table.
6. The image quality should be improved.
7. Proofread the paper once again
Reviewer 2 Report
The authors analyze the influencing factors of consumers’ e-waste recycling behavior through an SEM. The research content is clear, but there are some questions that need to be answered. More detailed comments are as follows:
Introduction
1) The innovation of this study should be stated more clearly. There are already many similar types of research. What is new in this one?
Literature Review
1) Hypotheses and model building should be put in the Methods chapter.
2) Why do PBC and PC not impact the ERB directly?
3) The content of the literature review is complex and needs to be condensed.
Data and Methods
1) Will the sample size be enough? It should be stated.
2) What is the sample select principle? Why do the authors only consider respondents with a willingness to recycle?
Results
1) The authors do not give reasons why Hypothesis 1 is inconsistent with other studies.
2) Most of the hypotheses of the MGA analysis failed to pass the test, indicating that population heterogeneity is associated with a low correlation to recycling. The authors should give explanations and comparisons with others’ studies.
Reviewer 3 Report
The paper deals with a very interesting topic and overall the research is well written.
My major concern is about the generalisability of the research; 159 questionnaires collected are not enough to generalisable Malaysia population in terms of age, gender, education etc. as the authors state. In my opinion such issue represents a critical aspect that undermine the publication of the paper. Considering that many of the relationships tested in the model proposed by the authors were tested by many other papers with better generalisability I think that the contribution of the paper could not be much relevant in this academic field.
On a minor note, even if the the introduction is well written and frame correctly the contribution in current relevant research on the topic, this part could be improved better underlining the research gaps in current academic researches and the novelty of the study.
I am sorry that my review is maybe not as positive as you would have hoped it to be. I hope that my input and suggestions on how to improve the paper are going to be helpful.
Round 2
Reviewer 2 Report
The authors analyze the influencing factors of consumers’ e-waste recycling behavior through an SEM. The research content is clear, and the author solved most of the problems after revising the article. Therefore, I think this article could be published in sustainability.
Author Response
Dear Prof./ Assoc. Prof/ Dr./ Mr./Mrs./ Ms.,
Thank you for your valuable comments and suggestions to improve our manuscript.
Reviewer 3 Report
My comments were addressed
Author Response

(The authors gave the same response as above.)
